# An Effective Shrinkage Control Method for Tooth Profile Accuracy Improvement of Micro-Injection-Molded Small-Module Plastic Gears

**DOI:** 10.3390/polym14153114

**Published:** 2022-07-30

**Authors:** Wangqing Wu, Xiansong He, Binbin Li, Zhiying Shan

**Affiliations:** 1State Key Laboratory of High Performance Complex Manufacturing, Central South University, Changsha 410083, China; csuxshe@csu.edu.cn; 2School of Mechanical and Electrical Engineering, Central South University, Changsha 410083, China; csulbb@csu.edu.cn

**Keywords:** micro-injection molding, small-module gear, plastic gear, non-linear shrinkage, tooth profile accuracy

## Abstract

An effective method to control the non-linear shrinkage of micro-injection molded small-module plastic gears by combining multi-objective optimization with Moldflow simulation is proposed. The accuracy of the simulation model was verified in a micro-injection molding experiment using reference process parameters. The maximum shrinkage ($Y_{1}$), volume shrinkage ($Y_{2}$), addendum diameter shrinkage ($Y_{3}$), and root circle diameter shrinkage ($Y_{4}$) were utilized as optimization objectives to characterize the non-linear shrinkage of the studied gear. An analysis of the relationship between key process parameters and the optimization objectives was undertaken using a second-order response surface model (RSM-Quadratic). Finally, multi-objective optimization was carried out using the non-dominated sorting genetic algorithm-II (NSGA-II). The error rates for the key shrinkage dimensions were all below 2%. The simulation results showed that the gear shrinkage variables, $Y_{1}$, $Y_{2}$, $Y_{3}$, and $Y_{4}$, were reduced by 5.60%, 8.23%, 11.71%, and 11.39%, respectively. Moreover, the tooth profile inclination deviation ($f_{H\alpha T}$), the profile deviation ($f_{f\alpha T}$), and the total tooth profile deviation ($F_{\alpha T}$) were reduced by 47.57%, 23.43%, and 49.96%, respectively. Consequently, the proposed method has considerable potential for application in the high-precision and high-efficiency manufacture of small-module plastic gears.

## 1. Introduction

In recent decades, with the rapid development of high-performance engineering plastics [1,2], small-module plastic gear has been widely applied in cutting-edge fields, such as aerospace [3], medical devices [4], and automobiles [5]. Compared with small-module metal gear, small-module plastic gear has obvious advantages, such as self-lubrication, strong shock and vibration resistance, and low noise [6,7]. Micro-injection molding is likely to become the main processing method for small-module plastic gear in the future because it enables production in large quantities [8,9]. However, because of the high shrinkage of engineering plastics and the irregular surface shape of gear teeth, plastic gear can undergo significant non-linear shrinkage during micro-injection molding [10,11,12,13]. Moreover, the specific surface area of small module gear teeth is small, which leads to complex and changeable interface heat transfer events during the cooling process [14,15]. Hence, the non-linear shrinkage of micro-injection molded small-module plastic gears is difficult to control.

Because of the small size of the small-module gear itself, the non-linear shrinkage phenomenon during the micro-injection molding greatly influences the tooth profile accuracy of small-module plastic gears [16]. In addition to the rational design of the mold structure and the selection of materials, the scientific selection of injection molding process parameters is an effective way to control shrinkage. There are many process parameters that affect the molding quality of small-module plastic gears. There is a complex, time-varying, nonlinear and strong coupling relationship between process the parameters and molding quality goals [17,18]. For this reason, it is particularly difficult to find a reasonable set of optimal process parameters. The traditional optimization method involves the constant adjustment of process parameters by injection molding personnel, based on experience, to obtain ideal parameter combinations. However, the traditional method is unable to achieve global optimal parameters, which greatly prolongs trial time, increases trial cost, and wastes raw materials [19,20]. Therefore, determining how to quickly obtain the optimal process parameters for micro-injection molding on a scientific basis is of great practical significance for the high-precision and high-efficiency manufacture of small-module plastic gears. 

The problem highlighted has attracted the attention of many researchers. In recent years, computer-aided engineering (CAE) technology and optimization design theory have been continuously developed [21,22]. CAE simulation technology has been used to replace the traditional continuous mode test method [23]. Combined with optimal design theory, it has been used in the optimization of injection molding process conditions, not only to improve the quality of plastic parts, but also to shorten the molding cycle [24]. Hakimian et al. simulated and analyzed the micro-injection molding process of 0.6 mm- tooth-thickness micro-gear based on Moldflow and optimized the process of maximum warpage and shrinkage using the Taguchi method [25]. Combining grey relational analysis and simulation analysis, Mehat et al. studied the influence of process parameters on the warping deformation of plastic gears [26]. 

In general, most current research considers warpage deformation and maximum shrinkage to be the process optimization goals of the gear when applying simulation models [25,26,27,28]. It is difficult to directly assess the degree of optimization of gear tooth profile accuracy. Optimization results deviate significantly from those observed in actual industrial production if optimization is based on simulation which is unverified by experiments. In many studies, the test samples are mainly subjected to orthogonal tests [15,25,29,30]. However, in orthogonal tests, problems of sample point accumulation can occur, and the test samples may not adequately represent all the important variables. Therefore, it is necessary to combine simulation with experiment, and to use CAE simulation and multi-objective optimization to control the non-linear shrinkage of small-module plastic gears. In addition, it is impossible to directly characterize and optimize tooth profile accuracy through simulation results. Therefore, it is important to identify optimization objectives that can be easily characterized and which fully reflect tooth profile accuracy. 

This paper focuses on the challenge of low tooth profile accuracy caused by non-linear shrinkage associated with the micro-injection molding of small-module plastic gears. We propose a novel method for shrinkage control that combines multi-objective optimization with Moldflow simulation to improve tooth profile accuracy, as shown in Figure 1. Local shrinkage, overall shrinkage, and key dimension shrinkage of small-module plastic gears are the main factors affecting tooth profile accuracy [15]. We use the maximum shrinkage ($Y_{1}$), volume shrinkage ($Y_{3}$), addendum diameter shrinkage ($Y_{3}$), and root circle diameter shrinkage ($Y_{4}$) as the optimization objectives for characterizing the non-linear shrinkage of the studied gear. Approximation surrogate models representing the relationships between the process parameters and the optimization objectives were established. The four optimization objectives were optimized in a non-normalized approach using the non-dominated sorting genetic algorithm-II (NSGA-II). Finally, micro-injection molding experiments on the studied gears were carried out using reference and optimal process parameters. As a result, shrinkage of the studied gear was well controlled. Measurements of size and tooth profile accuracy indicated that key dimension shrinkage errors between the simulation and experimental tests were less than 2%. Moreover, tooth profile inclination deviation $f_{H\alpha T}$, tooth profile deviation $f_{f\alpha T}$ and total tooth profile deviation $F_{\alpha T}$ were reduced by 47.57%, 23.43%, and 49.96%, respectively. Consequently, the proposed method can inform the high-precision and high-efficiency manufacture of small-module plastic gears.

## 2. Materials and Methods

### 2.1. 3D model and Material of Studied Gear

In this study, small-module involute cylindrical gear (module = 0.5 mm) was studied. A three-dimensional (3D) model and the dimension parameters of the studied gear are shown in Figure 2. Polyformaldehyde (POM) is most commonly used for production of plastic gears. It is a thermoplastic crystalline polymer with high strength and good wear resistance. The studied gear was made from POM-100 P, produced by DuPont, US, which is the most used high-performance crystalline engineering plastic for small-module plastic gears [31,32,33]. The physical properties of the materials used are shown in Table 1.

### 2.2. Moldflow Simulation

Autodesk Moldflow Insight (AMI) 2021 software, US, [34,35] was used for the simulation of small-module plastic gear micro-injection molding. For small-module gears, the polymer filling process is extremely short, and the process cannot be observed visually, so it is difficult to characterize experimentally. However, the polymer flow molding process can be studied visually using Moldflow. To facilitate later extraction of the 3D shrinkage gear model, a built 3D model of the studied gear was imported into the Moldflow software using an .stp extension. The gating system of the simulation model was established according to the actual forming gating system. A double-layer mesh was used for meshing, and the global edge length was 0.15 mm. The mesh chord angle was 20 degrees. The minimum curvature size was 10% of the global size. The mesh model was optimized by reducing the aspect ratio through operations such as merging nodes and integrating nodes. The injection gate position was then set according to the actual micro-injection molding. The gear simulation model is shown in Figure 3.

The analysis type chosen for this study was “thermoplastic injection molding”. The analysis sequence was “Filling + Packing + Warpage”. Then, POM 100-P from DuPont was selected in the software material library. Parallel computing was specified to improve the computing speed.

### 2.3. Experimental Design

Studies have found that the local shrinkage, overall shrinkage and key dimension shrinkage of plastic gears directly affect the tooth accuracy [25]. For this reason, the maximum shrinkage ($Y_{1}$), volume shrinkage ($Y_{2}$), addendum diameter shrinkage ($Y_{3}$) and root circle diameter shrinkage ($Y_{4}$) were taken to be optimization objectives for multi-objective optimization. The maximum shrinkage ($Y_{1}$) and volume shrinkage ($Y_{2}$) were obtained directly from the simulation results. The key dimension shrinkage was also assessed to verify the accuracy of the simulation model. The 3D model of the gear after shrinkage was extracted through Moldflow simulation. The diameter of the addendum and root circle were measured by CAD software. The addendum diameter shrinkage ($Y_{3}$) and the root circle diameter shrinkage $\left(Y_{4}\right)\;$were calculated using Equations (1) and (2).
(1)$$Y_{3}=\frac{d_{a}-d_{a}^{'}}{d_{a}}\times 100$$
(2)$$Y_{4}=\frac{d_{f}-d_{f}^{'}}{d_{f}}\times 100$$
where $d_{a}$ and $d_{a}^{'}$ are the addendum circle diameters of the studied gear and the shrinkage gear, and $d_{f}$ and $d_{f}^{'}$ are the root circle diameters of the studied gear and the shrinkage gear.

Based on previous research and long-term practical experience in the production of small-module plastic gears, the mold temperature, melt temperature, packing pressure, packing time, and cooling time were selected as input factors. Compared to traditional single-factor experimental designs and orthogonal experimental designs, the Box–Behnken design (BBD) method facilitates the analysis of individual and interactional effects of parameters on responses with fewer experiments and lower computational cost [36,37]. Thus, the BBD method was utilized to quantitatively analyze the influence of key process parameters on the shrinkage of small-module plastic gear. The micro-injection molding process levels were selected according to the range of process parameters recommended by the Moldflow material library, and production experience, as shown in Table 2. The 46 groups of BBD simulation results are shown in Table 3. It was found that the optimal results for $Y_{1}$, $Y_{2}$, $Y_{3}\;$and $Y_{4}\;$appeared in 39, 13, 12 and 41 groups of trials, respectively. Therefore, the interaction between the objectives was substantial, and the four objectives could not be minimized at the same time through conventional single-objective optimization.

### 2.4. Approximate Surrogate Model 

An approximate surrogate model can effectively save time required for CAE simulation. If the accuracy of the established model is reliable, the model can be applied to predict the properties of unknown points in space. In this study, Kriging models [38], radial basis function (RBF) models [39], linear response surface models (RSM) [40], RSM-Quadratic, RSM-Cubic and RSM-Quartet models between process parameters, and optimization objectives, were established using the Isight platform, US [41,42]. 

To determine if a model is representative, it is necessary to verify the convergence accuracy of the model through error analysis. The method used to verify the accuracy of the surrogate model involves comparison of the approximation of the predicted value and the test sample. This study mainly used R-square ($R^{2}$) to evaluate the accuracy of the built surrogate model. The most suitable surrogate model was selected through error analysis to predict the shrinkage of the gear more accurately. Moreover, the approximate surrogate model was suitable for further multi-objective optimization to minimize the shrinkage of the small-module plastic gear.

### 2.5. Multi-Objective Optimization

Since practical engineering problems are complex and involve uncertain factors, most problems can be studied with multi-objective optimization strategies. Multi-objective optimization algorithms provide more satisfactory solutions than single-objective optimization methods [43,44]. Multi-objective optimization methods have attracted the attention of researchers in different disciplines and fields. Many population-based algorithms have been proposed which have been used to solve the multi-objective optimization problem (MOP) in engineering applications.

The improved non-dominated sorting genetic algorithm, NSGA-Ⅱ [45,46], has good exploration performance and an efficient sorting process. The principle of the NSGA-II algorithm is shown in Figure 4. The algorithm not only uses crowding degree and a crowding degree comparison operator to replace the fitness sharing strategy under the control of sharing radius, but also introduces an elite strategy to ensure that the best individual is not lost. Therefore, in this study, the NSGA-II multi-objective optimization algorithm was selected to optimize gear shrinkage. For the algorithm settings, the population side was set to 12, the number of generations was set to 20, and the number of samples was 240, which meets the recommended value of between 20 and 200. The crossover probability crossover rate was set to 0.9, the recommended value was 0.6~1, and the remaining settings were set by default.

The mathematical model of the process optimization problem of gear shrinkage in this study was expressed as Equation (3).
(3)$$\{\begin{array}{c} F=\min\left[Y_{1}\left(\mathrm{mm}\right),Y_{2}\left(\%\right),Y_{1}\left(\%\right),Y_{1}\left(\%\right)\right]\\ s.t\{\begin{array}{c} 60\leq A\leq 100\left({}^{\circ}C\right)\\ 190\leq B\leq 230\left({}^{\circ}C\right)\\ 60\leq C\leq 80\left(\mathrm{MPa}\right)\\ 1\leq D\leq 5\left(s\right)\\ 10\leq E\leq 30\left(s\right) \end{array} \end{array}$$

### 2.6. Experimental Verification of Simulation Model and Optimization Method

First, the reliability of the gear simulation model required to be verified through micro-injection molding pre-testing with reference process parameters. The self-built micro-injection molding experiment platform is shown in Figure 5a, and the mold structure is shown in Figure 5b. Figure 5c is the actual mold of the studied gear. According to the recommended process for assessing materials, the reference process parameters are shown in Table 4. The diameter of the addendum circle and root circle were measured using a digital super-depth microscope (KEYENCE VHX-1000C, US). The addendum diameter shrinkage ($Y_{3}\;$) and the root circle diameter shrinkage $\left(Y_{4}\;\right)\;$of the gear samples were calculated using Equations (1) and (2), respectively. The reliability of the simulation model was verified by comparing the key dimensions shrinkage between the simulation and experiment under reference process parameters.

The results of multi-objective optimization require to be verified, and injection molding experiments were carried out under the obtained optimal process parameters. The optimization results of the NSGA-II algorithm were verified by Moldflow simulation. The accuracy of the optimized method was verified by comparing the key dimensions shrinkage and the tooth profile accuracy of the gear from the micro-injection molding experiments under reference and optimal process parameters. The tooth profile accuracy was automatically measured using an FPG 3002B small-module gear measuring machine [47]. Tooth profile accuracy [48,49] includes the deviation of tooth profile inclination ($f_{H\alpha T}$), tooth profile deviation ($f_{f\alpha T}$) and total tooth profile deviation ($F_{\alpha T}$). The gear clamping and specific clamping method for the small-module gear are shown in Figure 6.

## 3. Results and Discussion

### 3.1. Simulation Model Accuracy 

Since the flow channel was injected symmetrically, the shrinkage of the two gears was basically the same. For this reason, we only selected a single gear for analysis. The shrinkage results of the studied gear obtained by Moldflow analysis are shown in Figure 7. From Figure 7a, it can be seen that the shrinkage of the gear was different at different positions. The tooth part of the gear shrank the most. During the cooling process, the cooling rate indicated by the gear teeth was very high due to the large temperature difference between the mold and the polymer melt. Accordingly, the gear teeth measurements indicated that the volume of the polymer molecular chain changed greatly. The degree of shrinkage near the addendum and the root circle of the studied gear was the largest, which is consistent with previous research findings [15,25]. In addition, the shrinkage of the part where the gear shrank the most was 0.4183 mm. The test results for the key dimensions of the gear obtained by simulation are shown in Figure 7b. The volume shrinkage of the gear obtained by simulation was 13.49% (Figure 7c).

The teeth morphology of the gear simple under reference process parameters was observed using a 200-× microscope, as shown in Figure 8a. It can be seen that the forming quality of the gear teeth was intact, so the gear samples could be adopted to verify the simulation model. The addendum and root circle diameters of the gear samples measured under the 50-× microscope are shown in Figure 8b. The key dimension shrinkage of the gear derived through simulation and experiment was calculated using Equations (1) and (2), as shown in Table 5. It was found that the errors between the experiment and the simulation were all less than 2 %. Therefore, the simulation model was able to accurately simulate the shrinkage of the studied gear.

### 3.2. Approximate Surrogate Models and Effect Analysis of Process Parameters

The data in Table 2 was input into Isight software. The Approximate module was used to build the approximate surrogate models. Since the input factors are different types of physical quantities, the adaptation type was selected as anisotropic. To select the approximate surrogate model with the highest accuracy, the accuracy of the established five typical approximate surrogate models was compared; the results are shown in Table 6. It was found that the accuracy of the RSM-Linear model was clearly lower, and the accuracy of the optimization objectives was less than 85%. This indicated that there was a strong non-linear relationship between the optimization objectives and the process parameters. It was found that that the accuracy of volume shrinkage ($Y_{2}$) in the Kriging models and the RBF models was below 90%. The RSM-Quadratic, RSM-Cubic, and RSM-Quartet models all achieved the required accuracy. However, compared with the RSM-Quadratic model, the accuracy improvement using the RSM-Cubic and RSM-Quartet models was not very significant; however, the computational complexity increased substantially. Accordingly, the RSM-Quadratic model was applied to represent the response relationship between the process parameters and the optimization objectives of the studied gear. The RSM-Quadratic coefficients of the optimization objectives are shown in Table 6.

Therefore, the following RSM-Quadratic model was employed to represent the relationship between the input factors and the responses:(4)$$f=a_{0}+\overset{k}{\underset{i=1}{\sum }}a_{i}x_{i}+\overset{k}{\underset{i=1}{\sum }}a_{ii}x_{i}^{2}+\sum^{\;}\overset{k}{\underset{i=j}{\sum }}a_{ij}x_{i}x_{j},i<j$$
where f is the output response, $a_{i}=0,1,\ldots ,k$ are coefficients of regression, and $x_{i},i=0,1,\ldots ,k\;$are input factors. The RSM-Quadratic coefficients of the optimization objectives are shown in Table 7.

The significance of the RSM-Quadratic coefficients was evaluated, as shown in Table 8. The *p*-value for $Y_{1},$ $Y_{2}$, $Y_{3}$, and $Y_{4}$, as a function of $D^{2}$, was 0.000. Therefore, besides *A*, *B*, *C* and *D*, there was a strong interaction between *D* and *D*. To simplify the RSM-Quadratic, only *A*, *B*, *C*, *D*, *E* and $D^{2}$ were considered in the regression equations for the optimization objectives. The regression equations for $Y_{1},$ $Y_{2}$, $Y_{3}$, and $Y_{4}$ are Equations (5)–(8).
(5)$$Y_{1}\left(\mathrm{mm}\right)=0.37925-0.00094A-0.00116B-0.00445C-0.01898D+0.00024D^{2}$$
(6)$$Y_{2}\left(\%\right)=33.06385-0.26211A-0.10297B-0.03879C-1.97750D+0.42343D^{2}\;$$
(7)$$Y_{3}\left(\%\right)=14.13387-0.02770A-004364B-0.00744C-0.81237D+0.07602D^{2}\;$$
(8)$$Y_{4}\left(\%\right)=11.46747-0.04402A-0.04514B-0.08270C-0.59975D+0.03973D^{2}\;$$

Figure 9 shows the error analysis results of the RSM-Quadratic of the Gaussian spatial correlation function. The abscissa is the predicted value of the RSM-Quadratic and the ordinate is the actual measured value. The blue horizontal line is the average response value of the model, and the white background indicates that the model has high accuracy. The black line with a slope of 1 represents the ideal model. The closer the red dot is to the black slash, the more accurate and representative the surrogate model is. The red dots in the figure are evenly distributed on both sides of the black line and are very close. This indicates that the fitting accuracy of the RSM-Quadratic established in this study was very high. Therefore, the RSM-Quadratic established in this study can be used to predict the shrinkage of the studied gear micro-injection molding. It will save time spent on multiple Moldflow simulation attempts.

Then, the main effect diagram and variance analysis were used to analyze the influence of the process parameters on the optimization objectives. The main effect diagram is shown in Figure 10. Figure 10a,c,d are the main effect diagrams of maximum shrinkage, addendum diameter shrinkage and root circle diameter shrinkage, respectively. It was found that there was a positive correlation between the mold temperature and the maximum shrinkage, the addendum diameter shrinkage and the root circle diameter shrinkage. However, there was a negative correlation between the melt temperature, packing pressure and packing pressure and the maximum shrinkage, addendum diameter shrinkage and root circle diameter shrinkage. The maximum shrinkage, addendum diameter shrinkage and root circle diameter shrinkage increased first and then decreased with increase in cooling time. Figure 10b is the main effect diagram of the volume shrinkage. It can be seen that the volume shrinkage of the studied gear first increased and then decreased with increase in the packing pressure.

Table 9 is the variance analysis table of the maximum shrinkage in which the contribution ratio of the mold temperature and the dwell time is shown to be 5.05% and 81.61%. Table 10 is the variance analysis table of volume shrinkage in which the contribution ratio of the melt temperature and packing time is shown to be 13.68% and 73.71%. Table 11 is the variance analysis table of the addendum diameter shrinkage in which the contribution ratio of the mold temperature and the packing time is shown to be 7.99% and 80.59%. Table 12 is the variance analysis table of root circle diameter shrinkage in which the contribution ratio of the mold temperature and the packing time is shown to be 9.81% and 75.44%. Therefore, it can be concluded that the packing pressure had the greatest influence on the non-linear shrinkage of the small-module plastic gears.

Then, the interactive effects of the micro-injection molding process parameters on the shrinkage of the small-module plastic gears was further investigated. The interaction effect plots for the maximum shrinkage ($Y_{1}$), volume shrinkage ($Y_{2}$), addendum diameter shrinkage ($Y_{3}$) and root circle diameter shrinkage ($Y_{4}$) were obtained, as shown in Figure 11. If the two straight lines are not parallel or intersect, there is an interaction effect between the process parameters; the greater the degree of non-parallelism, the stronger the interaction effect. It can be seen that there was a large interaction between the process parameters, so the optimal combination of process parameters cannot be determined using a traditional single factor design.

### 3.3. NSGA-II Algorithm Optimization Results

The optimal process parameters obtained by NSGA-II were a mold temperature of 61.44 °C, a melt temperature of 193.64 °C, a packing pressure of 79.84 MPa, a packing time of 4.48 s and a cooling time of 20 s. Figure 12 shows the simulation results under the optimal parameters. The maximum shrinkage of the studied gear was 0.1400 mm (Figure 12a), which was reduced by 5.60% compared to the shrinkage of the gear under the reference process parameters. The volume shrinkage of the studied gear was 12.38% (Figure 12b), which was reduced by 8.23% compared to the shrinkage of gear under the reference process parameters. The addendum diameter shrinkage ($Y_{3}$) and root circle diameter shrinkage ($Y_{4}$) calculated according to Figure 12c were 5.618% and 5.762%, which were reduced by 11.71% and 11.39%, respectively. Compared with the optimization objectives predicted by the approximate surrogate models, the relative errors were all less than 5%, as shown in Table 13.

### 3.4. Optimization Method Validation

The measurement results for the key dimension shrinkage of the gear samples in the micro-injection molding experiment under reference and optimal process parameters are shown in Figure 13a. It was found that the addendum diameter shrinkage ($Y_{3}$) and the root circle diameter shrinkage ($Y_{4}$) decreased by 9.44% and 9.91%, respectively. Moreover, the tooth profile inclination deviation ($f_{H\alpha T}$), tooth profile deviation ($f_{f\alpha T}$) and total tooth profile deviation ($F_{\alpha T}$) were reduced by 47.57%, 23.43%, and 49.96%, respectively (Figure 13b). Figure 13c shows the tooth profile deviation curve of the gear sample with the highest tooth profile accuracy under optimal process parameters. It can be seen that the deviation near the addendum circle and the root circle was larger, while the deviation near the graduation circle was smaller, which was also close to the simulation results. Consequently, the tooth profile accuracy of the small-module plastic gears could be significantly improved using the proposed method. 

## 4. Conclusions

In summary, this paper has proposed an effective method to improve the tooth profile accuracy of micro-injection molded small-module plastic gears. First, based on the Box–Behnken design method, a simulation of the micro-injection molding of the studied gear was carried out. Next, the response relationship between the process parameters and the optimization objectives was analyzed using the RSM-Quadratic. The optimization objectives were non-normalized optimized using the NSGA-II algorithm. Finally, an experimental analysis of the micro-injection molding of the studied gear was carried out under reference and optimal process parameters to verify the optimization method. The main conclusions of this study are as follows.
The shrinkage of the gear was different at different positions. The tooth part of the gear shrank the most. Moreover, the shrinkage degree near the addendum and the root circle of the small-module plastic gear was the largest. The relative errors of the key dimension shrinkage between the micro-injection molding experiment and simulation were all less than 2%.The factor that had the greatest influence on the shrinkage of the small-module plastic gears was the packing time, and there was a complex interaction between the process parameters. The accuracy of the RSM-Quadratic for the optimization objectives reached more than 92 %.The simulation results showed that the $Y_{1}$, $Y_{2}$, $Y_{3}$, and $Y_{4}$ values of the studied gear were reduced by 5.60%, 8.23%, 11.71%, and 11.39%, respectively. Moreover, the tooth profile inclination deviation ($f_{H\alpha T}$), tooth profile deviation ($f_{f\alpha T}$) and total tooth profile deviation ($F_{\alpha T}$) were reduced by 47.57%, 23.43%, and 49.96%, respectively.

## Figures and Tables

**Figure 1 polymers-14-03114-f001:**
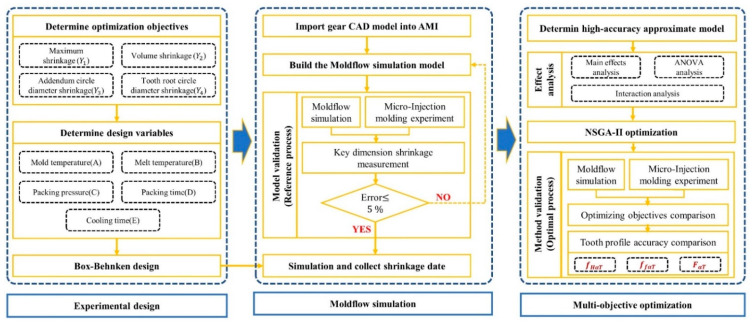
The tooth profile accuracy improvement method for micro-injection molded small-module plastic gears.

**Figure 2 polymers-14-03114-f002:**
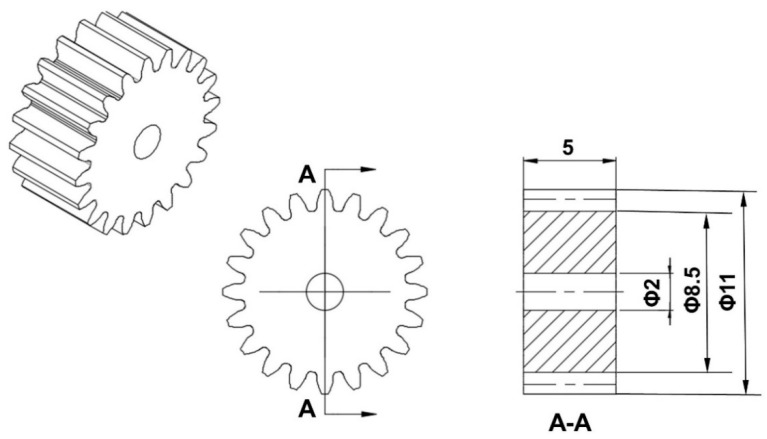
The 3D model and dimension parameters of the studied gear: Module (m) = 0.5 mm; number of teeth (z) = 20; modification coefficient (χ) = 0; pressure angle (α) = 20 °.

**Figure 3 polymers-14-03114-f003:**
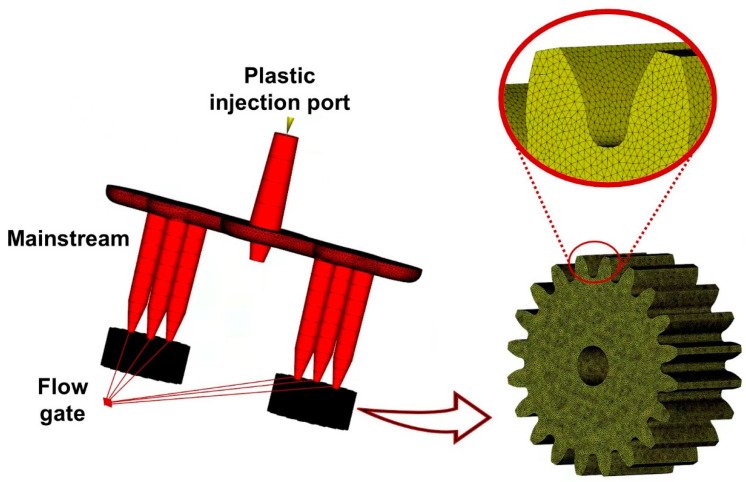
Simulation model of the studied gear.

**Figure 4 polymers-14-03114-f004:**
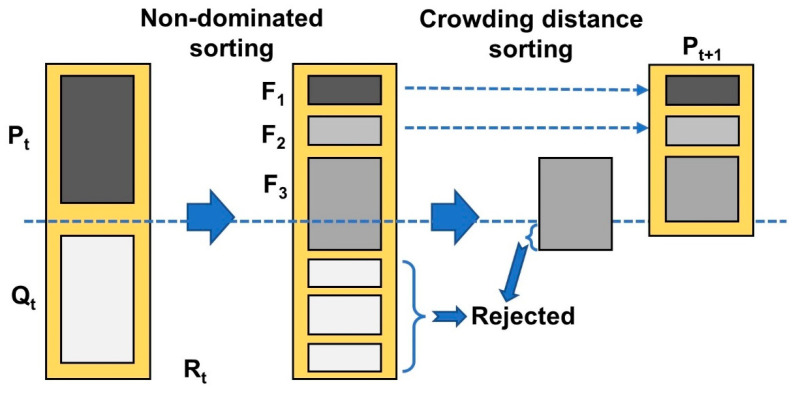
Schematic diagram of NSGA-II algorithm.

**Figure 5 polymers-14-03114-f005:**
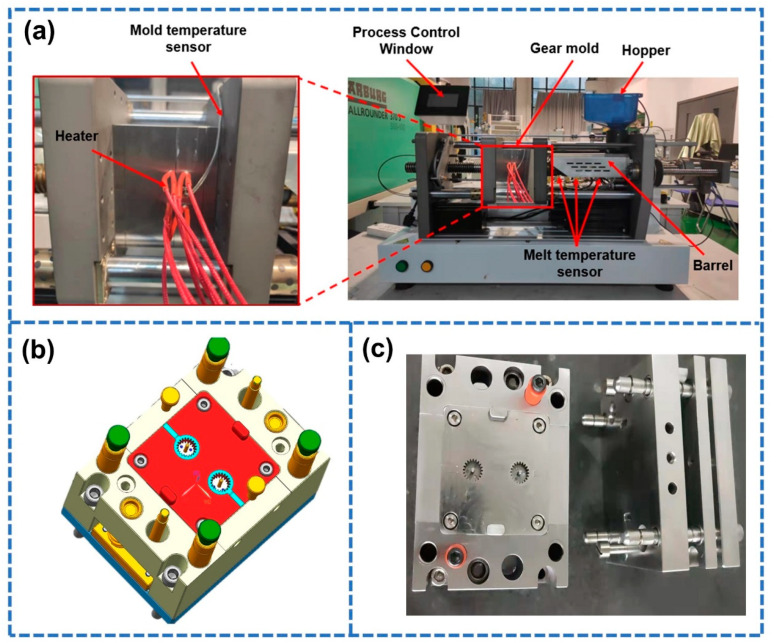
(**a**) Micro-injection molding experiment platform of small-module plastic gears. (**b**) Cavity layout of the mold. (**c**) The actual mold of the studied gear.

**Figure 6 polymers-14-03114-f006:**
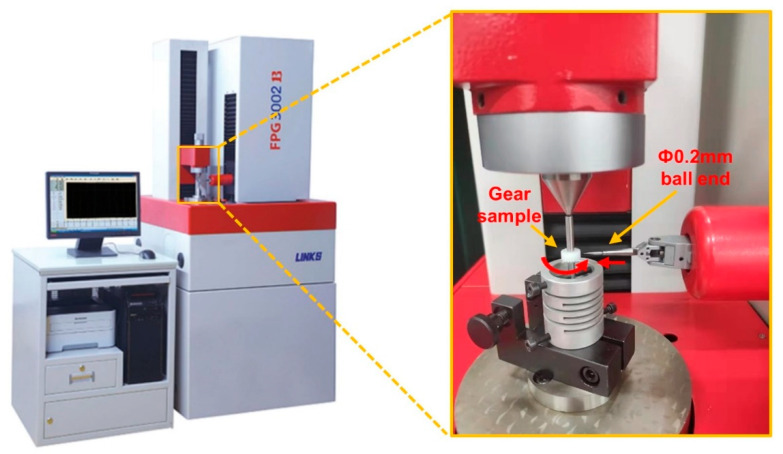
FPG 3002B small-module gear measuring machine and clamping method of small-module gear.

**Figure 7 polymers-14-03114-f007:**
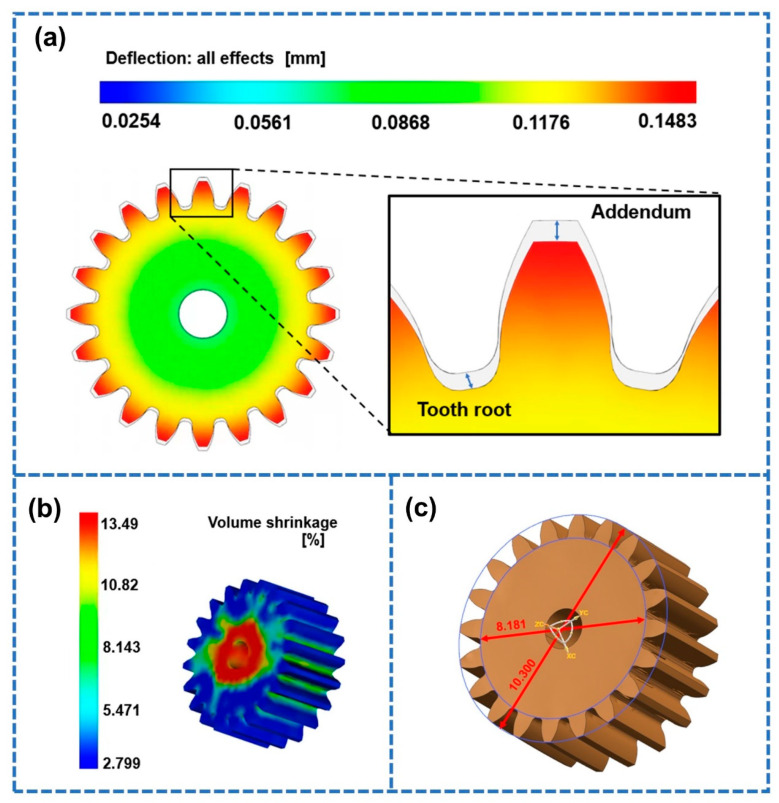
Simulation results under the reference parameters. (**a**) Shrinkage cloud diagram. (**b**) Volume shrinkage. (**c**) Addendum diameter and root circle diameter.

**Figure 8 polymers-14-03114-f008:**
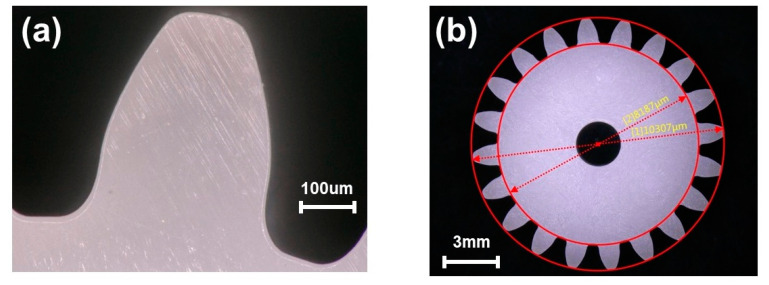
(**a**) The tooth morphology of the gear sample. (**b**) Key dimension results of experiment under reference process parameters.

**Figure 9 polymers-14-03114-f009:**
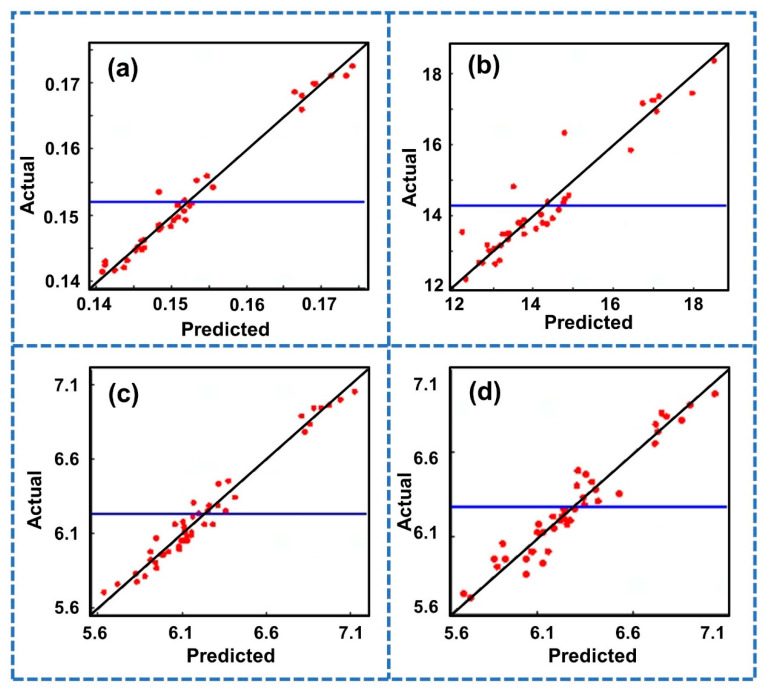
Residual analysis diagram of each response target of the RSM-Quadratic. (**a**) Maximum shrinkage (*Y*_1_). (**b**) Volume shrinkage (*Y*_2_). (**c**) Addendum diameter shrinkage (*Y*_3_). (**d**) Root circle diameter shrinkage (*Y*_4_).

**Figure 10 polymers-14-03114-f010:**
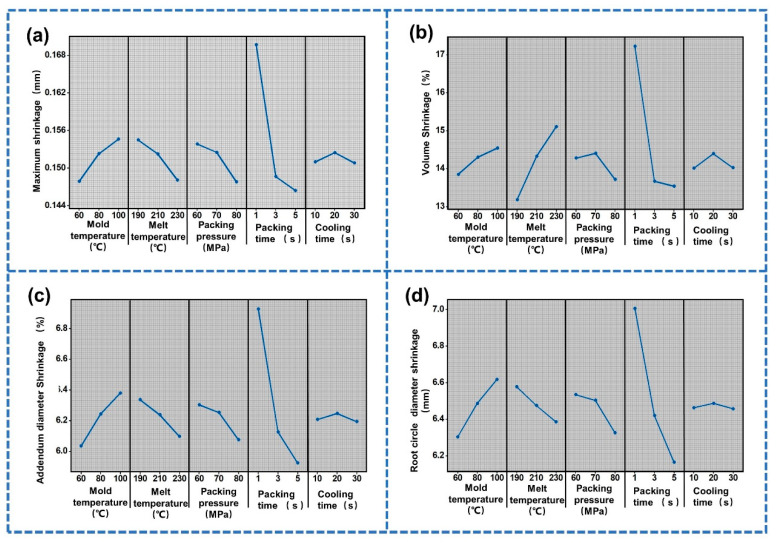
Main effect diagram: (**a**) Maximum shrinkage ($Y_{1}$). (**b**) Volume shrinkage ($Y_{2}$). (**c**) Addendum diameter shrinkage ($Y_{3}$). (**d**) Root circle diameter shrinkage ($Y_{4}$).

**Figure 11 polymers-14-03114-f011:**
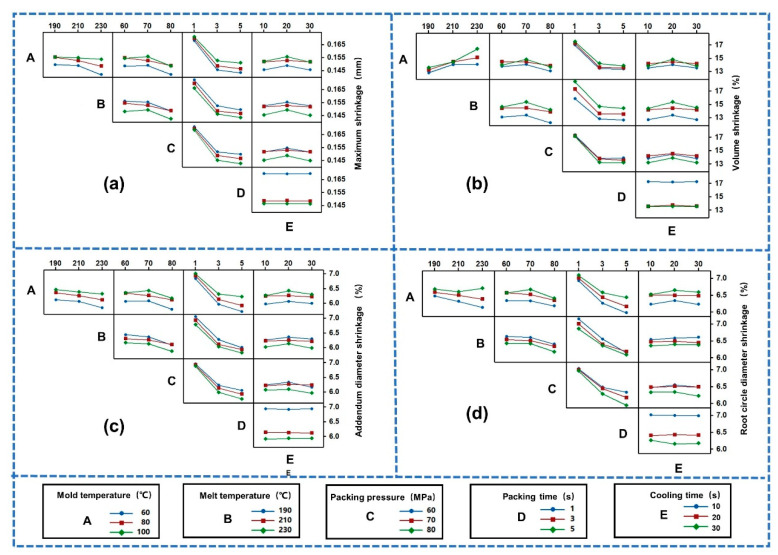
Interaction effect plot: (**a**) Maximum shrinkage ($Y_{1}$). (**b**) Volume shrinkage ($Y_{2}$). (**c**) Addendum diameter shrinkage ($Y_{3}$). (**d**) Root circle diameter shrinkage ($Y_{4}$).

**Figure 12 polymers-14-03114-f012:**
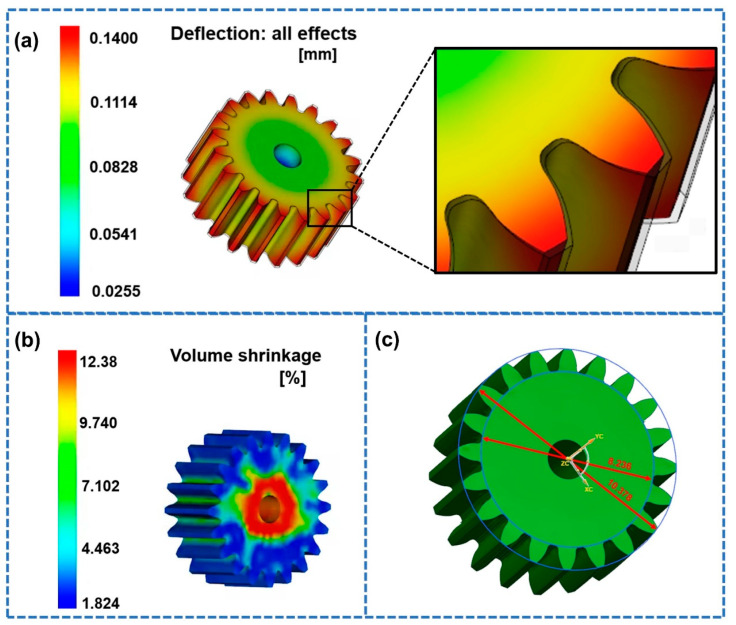
Simulation results under the optimal parameters. (**a**) Shrinkage cloud diagram. (**b**) Volume shrinkage. (**c**) Addendum diameter and root circle diameter.

**Figure 13 polymers-14-03114-f013:**
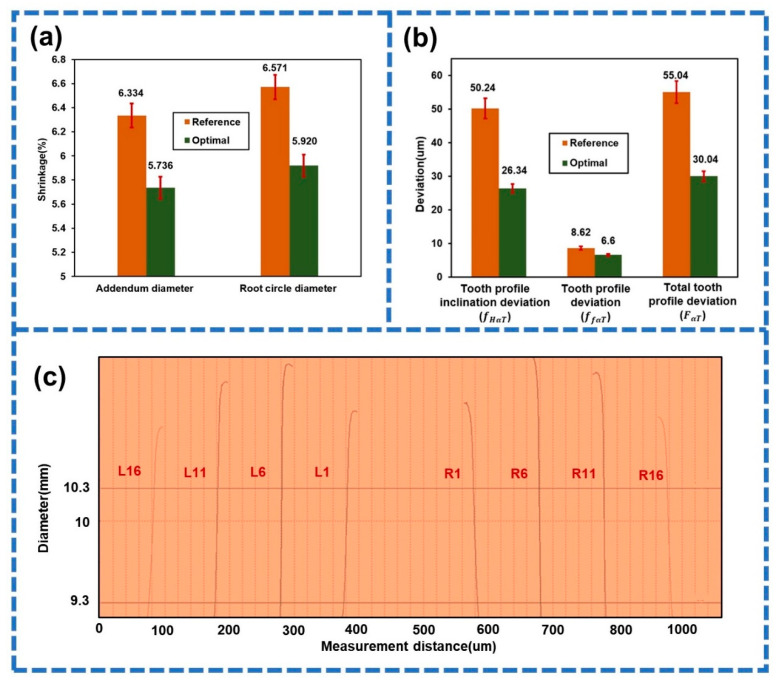
(**a**) Key dimensions’ shrinkage in the experiment under optimal process parameters. (**b**) Optimization results of tooth profile accuracy. (**c**) Tooth profile deviation curve of gear sample with the highest tooth profile accuracy under optimal parameters.

**Table 1 polymers-14-03114-t001:** Properties of DuPont POM-100 P.

Parameters	Test Method	Value
Water absorption (%)	ASTM D-570	0.25
Hardness (R)	ASTM D-785	120
Tensile strength (MPa)	ASTM D-638	69
Elasticity coefficient (MPa)	ASTM D-638	3360
Deflection coefficient (MPa)	ASTM D-790	3090
Shear strength (MPa)	ASTM D-732	66
$\mathrm{Tensile}\;\mathrm{impact}\;\mathrm{strength}\;(\mathrm{kJ}/m^{2}$)	ASTM D-1822	420
Thermal expansion coefficient (10^−5^ m /m°C)	ASTM D-696	12.2

**Table 2 polymers-14-03114-t002:** Factors and levels of key process parameters.

	Factors	Minimum Level	Intermediate Level	Maximum Level
** *A* **	Mold temperature (°C)	60	80	100
** *B* **	Melt temperature (°C)	190	210	230
** *C* **	Packing pressure (MPa)	60	70	80
** *D* **	Packing time (s)	1	3	5
** *E* **	Cooling time (s)	10	20	30

**Table 3 polymers-14-03114-t003:** Box–Behnken experimental design and simulation results.

Run#	Factor					Response of Optimize Objectives			
	*A* (°C)	*B* (°C)	*C* (MPa)	*D* (s)	*E* (s)	$Y_{1}$ (mm)	$Y_{2}$ (%)	$Y_{3}$ (%)	$Y_{4}$ (%)
1	80	210	80	3	10	0.1452	13.16	6.07273	6.33143
2	80	230	70	3	10	0.1452	14.37	6.01818	6.35429
3	80	210	70	3	20	0.1483	13.49	6.12727	6.40000
4	100	190	70	3	20	0.1553	13.55	6.45455	6.67429
5	80	230	60	3	20	0.1479	14.59	6.16364	6.42286
6	100	230	70	3	20	0.1536	16.35	6.30909	6.69714
7	80	210	80	1	20	0.1687	17.18	6.89091	6.97143
8	100	210	70	3	10	0.1519	13.78	6.25455	6.51429
9	60	210	70	3	30	0.1453	13.49	5.98182	6.21714
10	80	210	60	1	20	0.1711	17.38	6.96364	7.04000
11	80	190	70	1	20	0.1726	15.86	7.05455	7.15429
12	60	210	70	5	20	0.1431	13.33	5.70909	5.96571
13	80	190	80	3	20	0.1486	12.21	6.09091	6.40000
14	60	190	70	3	20	0.1494	12.75	6.10909	6.46857
15	80	190	70	3	30	0.1523	12.68	6.29091	6.60571
16	80	230	70	5	20	0.1433	14.40	5.81818	6.08000
17	60	210	70	1	20	0.1681	16.96	6.83636	6.92571
18	100	210	80	3	20	0.1484	13.64	6.16364	6.40000
19	80	210	70	1	30	0.1699	17.27	6.94545	6.99429
20	100	210	70	5	20	0.1507	13.81	6.21818	6.42286
21	80	210	70	3	20	0.1483	13.49	6.14545	6.46857
22	80	190	60	3	20	0.1560	13.08	6.43636	6.62857
23	100	210	70	3	30	0.1515	13.78	6.29091	6.58286
24	60	210	70	3	10	0.1453	13.49	5.96364	6.21714
25	80	210	80	3	30	0.1448	13.16	5.96364	6.21714
26	80	230	80	3	20	0.1422	14.18	5.87273	6.17143
27	80	210	60	3	10	0.1516	13.81	6.23636	6.46857
28	80	210	70	3	20	0.1483	13.49	6.05455	6.37714
29	80	210	60	3	30	0.1516	13.81	6.16364	6.49143
30	80	210	70	3	20	0.1483	13.49	6.10909	6.44571
31	80	190	70	3	10	0.1523	12.68	6.25455	6.53714
32	80	210	70	3	20	0.1483	13.49	6.18182	6.46857
33	80	210	60	5	20	0.1498	13.89	6.05455	6.33143
34	80	210	70	5	10	0.1463	13.51	5.90909	6.26286
35	100	210	70	1	20	0.1711	17.47	7.00000	7.08571
36	80	230	70	3	30	0.1449	14.49	5.98182	6.37714
37	80	210	70	3	20	0.1483	14.83	6.07273	6.46857
38	80	190	70	5	20	0.1493	12.65	6.00000	6.14857
39	60	230	70	3	20	0.1416	14.04	5.83636	6.12571
40	60	210	80	3	20	0.1418	13.02	5.78182	6.17143
41	80	210	80	5	20	0.1426	13.18	5.76364	5.94286
42	80	210	70	5	30	0.1462	13.51	5.92727	6.17143
43	100	210	60	3	20	0.1543	13.93	6.34545	6.56000
44	80	210	70	1	10	0.1699	17.27	6.94545	7.01714
45	60	210	60	3	20	0.1483	13.72	6.05455	6.33143
46	80	230	70	1	20	0.1660	18.39	6.78182	6.85714

**Table 4 polymers-14-03114-t004:** Reference process parameters.

Parameters	Value
Mold temperature (°C)	80
Melt temperature (°C)	210
Packing pressure (MPa)	70
Packing time (s)	3
Cooling time (s)	20

**Table 5 polymers-14-03114-t005:** Comparison of key dimension shrinkage rates obtained by simulation and experiment.

	$Y_{3}\;(\%)$	$Y_{4}\;(\%)$
Simulation	6.363	6.503
Experiment	6.334	6.571
Error (%)	0.300	1.035

**Table 6 polymers-14-03114-t006:** The accuracy of the Kriging models, RBF models, and the RSM-Linear, RSM-Quadratic, RSM-Cubic and RSM-Quartet models.

Approximation Surrogate Model	Accuracy (%)			
	$Y_{1}$	$Y_{2}$	$Y_{3}$	$Y_{4}$
Kriging	94.734	81.693	93.985	90.850
RBF	94.638	87.661	95.089	91.870
RSM-Linear	67.662	52.482	74.930	80.850
RSM-Quadratic	98.229	92.838	98.031	94.562
RSM-Cubic	98.151	92.037	97.856	95.086
RSM-Quartet	98.015	91.609	97.660	94.590

**Table 7 polymers-14-03114-t007:** RSM-Quadratic coefficient table.

Polynomial Term	$Y_{1}$	$Y_{2}$	$Y_{3}$	$Y_{4}$
Constant	0.37925	33.06385	14.13387	11.46747
A	−0.00094	−0.26211	−0.02770	−0.04402
B	−0.00116	−0.10297	−0.04364	−0.04514
C	−0.00445	0.03879	−0.00744	0.08270
D	−0.01898	−1.97750	−0.81237	−0.59975
E	0.00018	0.01842	0.02115	0.03480
$A^{2}$	1.12500	0.00029	0.00003	−0.00002
$B^{2}$	1.29167	0.00011	0.00006	0.00005
$C^{2}$	−1.16667	−0.00150	−0.00012	−0.00049
$D^{2}$	0.00024	0.42343	0.07602	0.03973
$E^{2}$	0.00000	−0.00123	2.66667	0.00009
AB	0.00000	0.00094	0.00008	0.00023
AC	0.00000	0.00051	0.00011	−0.00003
AD	0.00000	−0.00019	0.00216	0.00186
AE	0.00000	0.00000	0.00002	0.00009
BC	0.00000	0.00057	0.00007	−0.00003
BD	0.00000	−0.00487	0.00057	0.00143
BE	0.00000	0.00015	−0.00009	−0.00006
CD	0.00000	−0.00638	−0.00273	−0.00399
CE	0.00000	−3.73429	−0.00009	−0.00034
DE	0.00000	−2.61961	0.00023	−0.00086

**Table 8 polymers-14-03114-t008:** The estimated regression coefficients for $Y_{1}$, $Y_{2}$, $Y_{3}$, and $Y_{4}$.

Polynomial Term	*p*-Value			
	$Y_{1}$	$Y_{2}$	$Y_{3}$	$Y_{4}$
Constant	0.000	0.000	0.000	0.000
A	0.000	0.001	0.000	0.000
B	0.000	0.000	0.000	0.000
C	0.000	0.006	0.000	0.000
D	0.000	0.000	0.000	0.000
E	0.736	0.936	0.567	0.837
$A^{2}$	0.143	0.371	0.468	0.649
$B^{2}$	0.095	0.726	0.124	0.292
$C^{2}$	0.699	0.245	0.440	0.015
$D^{2}$	0.000	0.000	0.000	0.000
$E^{2}$	0.978	0.338	0.869	0.649
AB	0.002	0.053	0.188	0.003
AC	0.736	0.586	0.343	1.000
AD	0.015	0.968	0.001	0.012
AE	0.822	1.000	0.848	0.538
BC	0.343	0.541	0.567	0.837
BD	0.736	0.304	0.343	0.048
BE	0.866	0.873	0.447	0.681
CD	0.012	0.499	0.029	0.007
CE	0.822	1.000	0.702	0.223
DE	0.955	1.000	0.848	0.538

**Table 9 polymers-14-03114-t009:** ANOVA for maximum shrinkage.

Source	DF	Sum of Squares	Mean Square	F-Value	*p*-Value	Contribution (%)
A	2	0.000183	0.000092	78.80	0.000	5.05
B	2	0.000166	0.000083	71.14	0.000	4.58
C	2	0.000146	0.000073	62.71	0.000	4.03
D	2	0.002958	0.001479	1271.16	0.000	81.62
E	2	0.000000	0.000000	0.04	0.962	0.00
Error	35	0.000041	0.000001			
Total	45	0.003624				

**Table 10 polymers-14-03114-t010:** ANOVA for volume shrinkage.

Source	DF	Sum of Squares	Mean Square	F-Value	*p*-Value	Contribution (%)
A	2	2.012	1.0060	8.12	0.001	1.87
B	2	14.744	7.3719	59.53	0.000	13.68
C	2	1.450	0.7248	5.85	0.006	1.35
D	2	79.427	39.7137	320.71	0.000	73.71
E	2	0.133	0.0664	0.54	0.590	0.12
Error	35	4.334	0.1238			
Total	45	107.751				

**Table 11 polymers-14-03114-t011:** ANOVA for addendum diameter shrinkage.

Source	DF	Sum of Squares	Mean Square	F-Value	*p*-Value	Contribution (%)
A	2	0.47856	0.23928	77.67	0.000	7.99
B	2	0.23341	0.11671	37.88	0.000	3.90
C	2	0.20798	0.10399	33.75	0.000	3.47
D	2	4.82563	2.41281	783.21	0.000	80.59
E	2	0.00081	0.00040	0.13	0.878	0.01
Error	35	0.10782	0.00308			
Total	45	5.98814				

**Table 12 polymers-14-03114-t012:** ANOVA for root circle diameter shrinkage.

Source	DF	Sum of Squares	Mean Square	F-Value	*p*-Value	Contribution (%)
A	2	0.39574	0.19787	39.07	0.000	9.81
B	2	0.15007	0.07504	14.82	0.000	3.72
C	2	0.19460	0.09730	19.21	0.000	4.82
D	2	3.04317	1.52158	300.43	0.000	75.44
E	2	0.00077	0.00039	0.08	0.927	0.02
Error	35	0.10782	0.00308			
Total	45	4.03381				

**Table 13 polymers-14-03114-t013:** Simulation results and prediction results.

	$Y_{1}\;(\mathrm{mm})$	$Y_{2}\;(\%)$	$Y_{3}\;(\%)$	$Y_{4}\;(\%)$
**Prediction**	0.1409	11.92	5.631	5.885
**Actual**	0.1400	12.38	5.618	5.762
**Error (%)**	0.643	3.715	0.231	2.134

## Data Availability

Not applicable.

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
