# Peer review of "An Effective Shrinkage Control Method for Tooth Profile Accuracy Improvement of Micro-Injection-Molded Small-Module Plastic Gears"

_polymers, 2022, doi:10.3390/polym14153114_

Round 1
Reviewer 1 Report
The paper seeks to introduce an approach ‘’ An effective shrinkage control method for the tooth profile accuracy improvement of micro-injection molded small module plastic gears”. However, the authors should consider improving upon the quality to further highlight and emphasize.
1. Put “An” as a replacement for the initial word “A” at the beginning of the abstract to read “An effective” instead of “A effective”.
2. Put space between any variable and its respective unit. For instance, the abstract contains 47.57%, 23.43%, and 49.96% instead of 47.57 %, 23.43 %, and 49.96 % among others. consider correcting such anomalies throughout the manuscript. Notwithstanding these, most of the tables also contain these mistakes.
3. The introduction needs to be improved by relating to the mechanics of the studied materials and their mechanical characteristics. The references to be included are:
https://link.springer.com/article/10.1007/s00170-013-5300-7.https://link.springer.com/article/10.1007/s00542-019-04619-5.
4. Based on the understanding of what an abstract should contain, consider adding one or two lines highlighting the significance of the study at the end of the abstract.
5. The maximum number of words allowed in the keywords is three. For instance, you have a “hybrid composite flexible riser” which is more than the number of words allowed. Either reduce it or abbreviate it.
6. Some of the texts inside figure 1 are not visible at a distance glance. Change the font to a clearer font.
7. A unification of the writing style shall be followed. Authors sometimes type "Fig.", elsewhere "Figure" as in the caption text. Choose one format and keep it constant.
Reviewer 2 Report
Review of polymers-1834964
1. The execution and writing in this manuscript is quite OK. However, please add more content about POLYMERS in this manuscript, not just about statistical design of experiment and machining or manufacturing. The industrial engineering content is too dominant in this manuscript. The content about polymer in this manuscript is just the fact that it uses Dupont POM-100 P (stated only as “high-performance crystalline engineering plastic”).
2. Based on the Box-Behnken design provided in the Table 3, the results were reanalyzed by using Minitab. Enclosed is my analysis (using coded units). There is an overlooked result, where the D*D (interaction of D with D) is not discussed, while the P-value for y1, y2, y3, y4 as function of D*D is 0.000 (<0.05, or highly significant). Please WRITE THIS important statistical result in the text.
3. Moreover, the regression equations for y1, y2, y3, and y4 MUST ALSO BE DISPLAYED, as function of A, B, C, D, and DD. It is clearly observed that variable E (cooling time) is not significant for all y1, y2, y3, and y4, which is obviously visualized in the main effect plots (Figure 10a, 10b, 10c, 10d). However, variable D (packing time) is heavily significant (based on the visualization in Figure 10), as well as its own interaction D*D. Please add the regression equations of y1, y2, y3, and y4 as function of A, B, C, D, and DD.
-----Please download the review report. The long and exhaustive details of Minitab re-analysis are there-----
4. Line 47: Delete “And”. Please do not start a sentence with “And”.
5. Line 52: The word “blind” is not suitable for the context of this sentence. Please change with other word(s).
6. Line 90: Delete “And”. Please do not start a sentence with “And”.
7. Table 1: The unit of tensile impact strength is kJ/m2, with uppercase J.
8. Line 121: …in .stp extension.
9. Line 121: Delete “And”. Please do not start a sentence with “And”.
10. Line 126: …nodes. The injection gate position was then set according…
11. Heading of Table 3: Please write the unit of temperature as °C --> Not the “ready mix” ℃ --> Please use the correct unit that is shown in Table 2, row 2 and 3.
12. Line 185: …and full of…
13. Line 189: Delete “And”. Please do not start a sentence with “And”.
14. Line 267: Delete “And”. Please do not start a sentence with “And”.
15. Line 276: Delete “And”. Please do not start a sentence with “And”.
16. Line 392: Delete “And”. Please do not start a sentence with “And”.
17. References: Some are with DOI number, some are not. Please make them in uniform manner.
18. Reference 49: What is the journal name?
-----Again, please download the review report. The long and exhaustive details of Minitab re-analysis are there-----

Round 2
Reviewer 2 Report
Review of polymers-1834964-v2
The authors have put considerable effort to make this manuscript improved. This manuscript can be accepted for publication now.